# Keeping the Knives Sharp: Socioeconomic Innovation in the Artisan Sector of Butchery in Italy

Michele Filippo Fontefrancesco [1,2,*] and Andrea Costa [1]

[1] University of Gastronomic Sciences, 12042 Pollenzo, Italy
[2] Department of Anthropology, Durham University, DH1 3DE Durham, UK
[*] Correspondence: m.fontefrancesco@unisg.it

**Abstract:** This paper investigates the socioeconomic and cultural transformation in food artisan entrepreneurship due to the complexification of the food sector and ongoing globalisation through a case study conducted among the artisan butchers of Genoa, Italy. The butcher's trade has enjoyed centrality and social and cultural promotion that made butcher shops key places in the urban foodscape. However, this centrality is challenged by both new consumption trends and the imposition of large-scale organised distribution as the fulcrum of mass food trade. These changes raise the question about the future of the profession and its knowledge. This paper addresses this question by investigating the structure of the butcher's business and the practices involved in knowledge creation and transmission, exploring the factor of change and its effects on the butcher's profession, as well as the very foundational aspects of the artisanship. The research suggests that artisanship should be read as a form of entrepreneurship capable of placing and moving the craftsman within the global social hierarchy of a community. Therefore, the preservation of artisanship and its sociocultural complexity cannot be limited to the mere preservation of gastronomic forms and techniques.

**Keywords:** artisanship; food sector; meat sector; retail sector; Italy; ethnography





## 1. Introduction

In the past 20 years, the retail food sector has undergone substantial changes [1]. The global increased mobility of goods and people has led to an expansion of the array of foods available in the markets thanks to the introduction of new, foreign products sold alongside more traditional ones, as well as the invention of new foods combining different culinary traditions [2–4]. Moreover, while the 20th century marked the advent of supermarkets [5], recent decades have seen the success of new forms of online commerce [6]. All these new forms of retailing have furthered and strengthened the food global value chains, thereby transforming them [7–10]. On the one hand, they favoured a global standardisation of agrifood production and food consumption patterns [5]. On the other hand, they promoted industrial mass production, strengthening the sociocultural prestige of these goods [11]. Finally, both the birth of shopping malls in the outskirts of the urban centre and the success of e-groceries moved food purchase away from the traditional locations of the artisan shops in the city centres and showed the fragility of the traditional retail system based on small business and food artisan professions, such as bakers, greengrocers, and butchers [12,13]. Hence, the last 20 years were a period of strong transformation in food artisanship.

Artisanship is a widely debated concept. It is generally opposed to mechanical reproduction based on an ideal contraposition between a non-alienating form of production (crafts-based) and an alienated one (machine-based), and it has often been associated with concepts such as design, manual work, small production, originality, oral lore, and traditions [14]. Since the late 20th century, this concept has been increasingly used in the study of the food sector [15] to identify those small-scale productions mostly based on manual work [16–18]. Sennet indicates artisanship as "a special human condition of

being engaged" [19] that "suggests ways of using tools, organising bodily movements, thinking about materials that remain alternative, viable proposals on how to conduct life with skill" [19] against a model of life and production based on modern detachment of body and hand. This definition resonates with Ingold's indication of artisanship as a complete expression of all forms of knowledge underpinning human making [20], or Marchand's [21] reading of artisanship as an expression of embodied knowledge that underpins material manipulation. In this respect, the concept of artisanship refers also to food professions, specifically those that require the manipulation of raw materials to prepare and create new food products. While chefs achieved unprecedented popularity in recent decades [22,23], other professions did not enjoy the same success, being in the margin of a global hierarchy of value [24]. One of these is the profession of butchers.

Butchery is the process of cutting and preparing animal carcasses for consumption. This practice enjoyed particular prestige given the strong symbolic value of meat as a food marker of social status, prestige, and prowess [25,26]. The butcher's trade has enjoyed centrality and social and cultural promotion that made butcher shops key places in the urban foodscape. However, this centrality is challenged by both new consumption trends critical of meat consumption [27], and the imposition of large-scale organised distribution as the fulcrum of mass food trade [28]. These changes, which are reducing the profitability and possibility of continuing the business as it used to be, raise a question about the future of the profession and its knowledge—a question that is shared by other artisan food professions.

This paper addresses this question by investigating the sociocultural transformation that affects business in Italy through a case study [29] focused on the city of Genoa. Specifically, it investigates the structure of the butcher's business and the practices involved in knowledge creation and transmission, exploring the factor of change and its effects on the profession, as well as the very foundational aspects of the artisanship.

The article presents the methodology and the activities conducted during the research, then presents the specificities of the business in Genoa and its recent development. In so doing, it highlights the elements needed to further explore the characteristics of food artisan knowledge and its reaction to social and economic change.

## 2. Materials and Methods

The article is the result of a case study [29] conducted in Genoa between March and December 2022. Italy is one of the countries most renowned for its gastronomic tradition, a specificity that is the result of the strong regionalism and geographical diversity of the peninsula [30]. In Italian gastronomy, meat has played a crucial role as one of the key symbols of prosperity and affluence since the Middle Ages [31,32]. With it, butchers have played a key role in the food profession, a cultural prominence that reached its peak in the 1980s [33]. In the face of this success, the past decades have been marred by a decline caused by the transformation of the retail sector due to the proliferation of supermarkets, epidemiological occurrences, and a new attitude of consumers toward the consumption of meat [34]. The dynamics that affect Genoa are framed within this context.

Genoa is one of the 10 largest cities in Italy, with over 500,000 inhabitants. Since the Middle Ages, the economy of the city has been based on its port, one of the most important in the country [35]. During the 20th century, it met a fast industrialisation that made Genoa one of the three main poles of the so-call Economic Boom of the 1960–1970s [36]. Despite the de-industrialisation of the past 30 years, Genoa still represents one of the key economic centres in northwestern Italy and a thriving place of commerce.

As a result of its prosperity, Genoa has been a centre of important gastronomic creativity, capable of developing a distinct cuisine [37]. Unlike other port cities of which gastronomy focuses on the primary use of fish resources, Genoese cuisine is characterised by the substantial presence of meat dishes [38]. This refers to the use of red meat, mainly beef, as well as white meat, poultry, and small animals [39]. In this respect, butcher shops played a crucial role in the life of the city, as well as in the expression of its gastronomy.

They were concentrated in the area of Macelli di Soziglia (tr. Soziglia slaughterhouse), where the city's main slaughterhouse was located (Figure 1). The Macelli are in the middle of the historic centre, in the area commonly called the "Vicoli" (tr. the alleys, due to the narrowness and tortuosity of the streets). Beginning in the 13th century and for over 500 years, this area, and specifically nearby Piazza Soziglia (tr. Soziglia square) and via Macelli di Soziglia (tr. Soziglia slaughter houses alley), has housed the city's slaughter houses and butcher shops as well as the stables where cows, pigs, and other animals were kept [40]. Although in the past decades, the slaughterhouses closed down, new ones were built in more external areas of Genoa (e.g., Piazzale Bigny in the northern periphery of the city), and although the number of butcher shops has substantially reduced to a handful, the area is still associated with meat retailing by the locals.

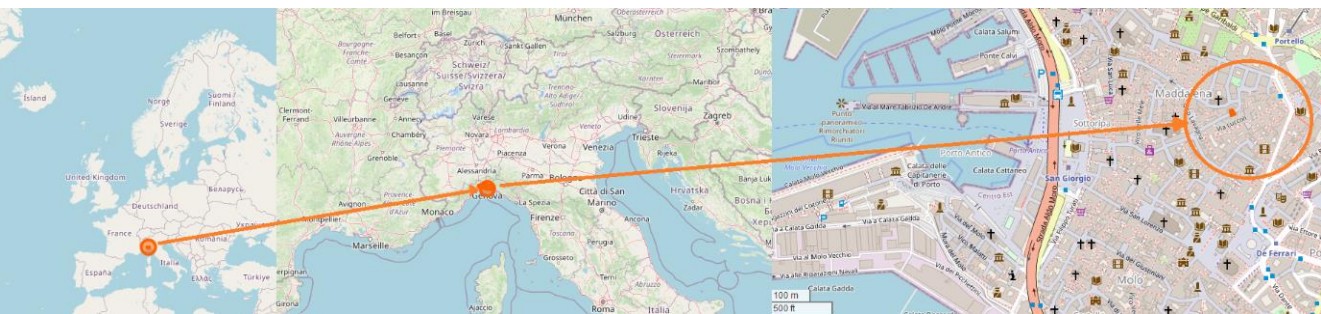

**Figure 1.** Localisation of the field (map based on OpenStreetMap cartography).

The research moved from this area and expanded to the city (Figure 2). Based on the methodologies of street ethnography [41] and business ethnography [42], the research focused on butcher shops in the city, exploring their operations and the dynamics that develop there. Following Healy et al. [43], the research conducted its analysis within the space of shops. Extensive in-depth interviews were conducted with the butchers running the businesses (Carey, 2013; Guala, 2003). Based on the life story approach [44], the interviews explored the modern history of butchery in the city, how the profession developed, and the changes in clientele.

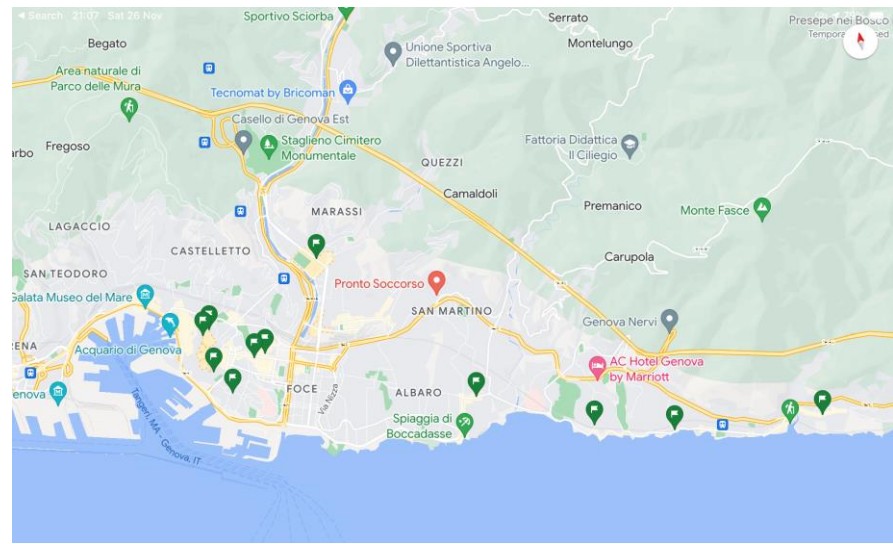

**Figure 2.** Localisation of the respondents (flagged in green) based on Google Map cartography.

The sample includes 13 professional butchers (Table 1) who own and manage at least one butcher shop. The sample was built starting from the butchers now active in the area of Macelli di Sozziglia and identifying others through snowball sampling [45], recognised

by respondents as relevant for research for their professional experience and knowledge. In this sense, the research has also extended beyond the original scope of research.

**Table 1.** Summary of the informants involved in the research.

| Identification | Age | Trainer | Education | Youth Entrepreneurship (under 25) | Family Business |
|---|---|---|---|---|---|
| Int. 1 | 20–34 | Kin-related (father) | Junior high school | | Yes |
| Int. 2 | Over 65 | Non-kin-related | Junior high school | | |
| Int. 3 RC | Over 65 | Kin-related (father) | Junior high school | | |
| Int. 4 | 35–65 | Kin-related (father) | Junior high school | Yes | Yes |
| Int. 5 ADC | 35–65 | Non-kin-related | Junior high school | Yes | |
| Int. 6 LS | 35–65 | Non-kin-related | Junior high school | | |
| Int. 7 EA | Over 65 | Non-kin-related | Junior high school | | |
| Int. 8 MT2 | 35–65 | Non-kin-related | High school | | |
| Int. 9 IS | 35–65 | Kin-related (father) | Junior high school | | Yes |
| Int. 10 CF | 35–65 | Non-kin-related | High school | | |
| Int. 11 DP | Under 35 | Non-kin-related | High school | | |
| Int. 12 RN | Over 65 | Kin-related (great-grandfather) | Junior high school | | Yes |
| Int. 13 VB | 35–65 | Kin-related (father)School of Italian Butchers training (1984–1986) | Junior high school | Yes | Yes |

The profile of the informants corresponds to what is commonly recognised as a professional craftsman who has reached the necessary professional skills to manage the main stages of production [19]. While the older informants (over 65) completed only their junior high school education (9), all the other informants completed a high school education (3). None have completed university studies. The training was completed through apprenticeship from older artisans (only one informant completed formal training in a school). The trainers were often kin-related (6), and this pattern is not significantly related to the age of the informants. All the informants are men, since no woman butcher owning and running a business was identified in Genoa.

The average age is over 50 years old. The sample includes two informants under 35 (in their late 20s), six informants between 35 and 65 years old, and five over 65. While the interviews show that until the 1990s it was not uncommon for people in their 20s to start their business (three informants shared this experience), the event is rare at the present, as portrayed by the last national census [46].

Field research was conducted by AC, under the supervision of MFF. A preliminary analysis of the data was produced by AC. Data were further analysed by MFF, who developed them in the present form of the paper.

## 3. Results

### 3.1. An Artisan Profession

Sennet [19] identifies the development of an artisan profession in a process of acquiring and testing creative skills and competence that leads the individual from apprenticeship to journeymanship, and from journeymanship to mastership. This is a path of social legitimation marked by specific rites of passage [47], but it is also and above all a process through which the individual embodies and practices knowledge [21]. This process is followed by the butchers in Genoa.

Butchery is a craft that is acquired by direct experience under the guidance of an older artisan. It is based on the knowledge that is "stolen with his eyes" (Int. 7) during everyday practice in the shop. All the butchers describe this process that starts with the completion

of the easiest task. As Int. 7 explains, "the first thing I had to learn is how they behaved and handled things". Int. 12 remembers: "It was in the '70s when I began working in a butcher's shop. I began because I was passionate and wanted to undertake this profession. So, I got the first opportunity I had to learn from a butcher. The first thing he taught me was to clean the shop and all the tools".

After managing the tools, the apprentice starts learning how to use them. The first cut that an apprentice learns to deal with is the belly. Int. 4 explains: "The first thing you learn is how to debone and skin the belly of the cow, as this allowed apprentices to make more mistakes at the beginning as that cut isn't as fine as other fresh cuts, such as the posterior like the thigh". Similarly, Int. 7 indicates: "The belly is one the least refined cuts of the cow; therefore, you can allow yourself to make more mistakes. It is perfect for learning. Then, you worked from the belly up to the most refined cuts of the cow". Another easy preparation is linked with the so-called "prontocuoci" (ready cuts, such as hamburgers, roulades, skewers, or other preparation-ready and fast-to-cook cuts). As Int. 10 explains, "I wasn't working, and a friend of his asked me to help him out in his shop. He was my first teacher. I had been a baker; therefore, I was used to doing things manually. My friend understood it and made me do prontocuoci as a first assignment. That's how I learned".

Learning takes place in the workshop, experiencing the everyday reality of the shop (only Int. 13 started his activity with participation in a vocational training course). Learning passes from the relationship with the elder butcher who becomes master and "almost father" (Int. 1) for the apprentice. In this sense, it is an all-encompassing experience which builds a strong personal and professional identity. Moreover, among those for whom butchery is a family occupation, the trade is permeated by a sense of legacy that is proudly displayed once the history of their life and business is told. This is the case for Int. 12:

"I did not choose to be a butcher. It was a stroke of unluck. I am joking. I was supposed to continue studying. My great-grandfather opened the shop over 110 years ago. When I was a boy, I was studying and, in the meantime, my grandfather was teaching me the profession. My dad sadly passed away, and I had to start working under my grandfather in the shop. The passion came along. After all, I really did like the profession; nonetheless, I was studying. I was a quick learner. As they would leave me alone in the shop very often, I had to manage the store and the clients all by myself. This is because I learned very quickly. We have this trade in the blood. My son now continues the activity after I officially retired some years ago."

Entering the profession is linked to a sense of cultural intimacy [48] dictated by kinship or by the personal knowledge of the master butcher. Becoming a butcher is the result of a direct interpersonal relationship between trainer and trainee which is not mediated by other institutions, such as a school. As Int. 2 puts it, "It does not happen to become a butcher. Either you have it in your blood or you fall in love because someone makes you live it [butchery] before you even set foot in the shop". The apprenticeship lasts several years and is completed when the butcher knows how to manage the entire carcass of the animal and prepare all the cuts from it. However, in the words of my informants, it is only official when the butcher becomes able to also manage the shop and its clientele, thus being able to run the commercial activity profitably. As put by Int. 3, "In the shop, we all [butchers of Genoa] always had some apprentices. The most talented and enterprising either took over the shop or opened their own and, at some point, started teaching a new generation the craft. The others remained to work as employees, giving strength to the business".

The company structure of the individual butcher shop followed a usual model that is at the same time spatial, occupational, and gendered. Similarly to what was highlighted by Simpson et al. [49], it hinges on the presupposition that a butcher should be a man and develops as a family business that involves husband (butcher) and wife, as well as the next generations (children, grandchildren), especially when they are boys. The shop accordingly develops a precise construction of the space that distinguishes between the shop floor and the back shop. The latter includes laboratories and warehouses. The shop floor is marked by the presence of a meat counter, on which meat is displayed, and of a checkout. The

space for the butcher and employees is located behind these structures, and from there they serve the customers, who are in front of them. The space behind the meat counter as well as the back shop are managed by the butcher and his men assistants, whereas the checkout is run by women, who are also in charge of bookkeeping. This structure is attested in the memory of the informants and, in some cases, is still what occurs in the present day. It was also the model reproduced when a new butcher shop opened, still based on the centrality of the butcher-shop owner around which a community of practitioners orbited, making the shop run. However, the informants highlight the impossibility or difficulty of maintaining the employment levels of the past, the failing intergenerational presence, and the shrinking of their businesses and, above all, the sector. "We were many, and nowadays we are few butcher shops. People buy less meat, and supermarkets are taking away shares of our market and the sector have to change to keep going," commented Int. 9, "We and the sector have to change to keep going . . . ".

*3.2. Changes in Consumption Pattern*

During the 20th century, meat consumption experienced a radical transformation. In the first half of the century, meat, especially red meat, was rarely found on the tables of the popular classes [50]. Meat was mostly food for special occasions and festivity, consumed sparingly and with the utmost care to avoid waste, also through the reuse of leftovers in preparations such as sauces, meatballs, stuffing, and fried food. For the popular classes, access to red meat was limited to the use of second-grade cuts, such as muscle, tail, or offal, and often of older animals. They were cuts that required culinary care and long cooking to cope with the strong flavour and the fibrous and tough consistency.

In a social context in which it was common for women, once married, to play the role of family caregivers under whose responsibility the purchase of food was placed, women had a central role in the choice as well as in the purchase and preparation of meat. The informants emphasise the role of women, *massaie* (tr. Housewife). "They would handle all the house chores along with grocery shopping, and they were the ones that would pick meat cuts and cook for the family", explains Int. 2, "Massaie were a fundamental part of the clientele still in the '60s. They had the knowledge of meat cuts, and unlike the clientele of today, they wouldn't linger on noble cuts; rather, they would pick out anterior cuts which would require long cooking times, as the anterior part of the animal is the most muscular one". This is due to the morphology of the cow, as it rests the most on its anterior paws, causing these cuts to be harder texture-wise. Today, it is unlikely that younger customers would order these types of cuts.

Behind this choice, the interviews identify two main motivations: on the one hand, the preference for simpler and faster cuts to prepare, and on the other hand, the lack of gastronomic knowledge among modern consumers, as well as the facilities and time available to prepare the cuts. The butchers indicate the change occurred starting in the 1980s.

Spinelli [33] describes how the 1980s in Italy marked a radical change in consumer food habits. Italians appeared to fall in love with new industrial products and especially with cooking preparations and dishes closer to those of American tables seen in movies or telefilms. Moreover, in the same decade, the Italian family structure transformed. Specifically, the decade marked the decline of the single-income family model in favour of realities in which both partners or consorts actively participated in the job market [51]. This led to the disappearance of someone who could care for long food preparations every day. Consequentially, a new demand for food based on preparations or cuts that are quick to prepare became dominant.

Butchers had to change their product, expanding the range of products to include more of "delicacies". "The '80s saw an increase in requests for 'prontocuoci', and we butchers needed to adapt and keep up with the customers' requests", remembers LS, who owned a butcher shop that also served delicatessen such as filled zucchini flowers, octopus, potatoes, and Russian salad back in the 1980s. "There was a demand for more elaborate proposals, not just the classic breaded slices 'Milanesi' [beefsteak], but rather cuts such

as tartare seasoned with different condiments, stew, and many more". In the face of this change, in a few years, informants also noticed a progressive thinning of the gastronomic knowledge of customers, now not only women. As explained by Int. 6: "If I compare my clients born in the '70s and their parents … the younger seems to be the least interested in cooking I have ever met … on a general basis they can barely distinguish a slice of liver from a beefsteak". However, in the face of this rarefaction, over the past 15 years, the increased media attention, described by Corvo [1], has brought new consumers closer to new recipes, traditional cuts disregarded in the wake of the 1980s consumption change, and more complex preparations, imposing a new global imagination of meat cuisine. As put by Int. 6:

"There is a difference between women born in the '70s who experienced the event of globalisation and rise of fast foods compared to a woman born at the beginning of the '90s. On a general basis, the people born in the '70s due to the event of industrialisation and the rise of fast food may have less knowledge in regards to cooking, whereas 35-year-olds seem to have more interest in regards to cooking. These clients, when they come to my shop, seem to have more knowledge and more curiosity about which cuts serve which dish compared to 50-year-olds. I believe it is also due to the rise of mass media engaging customers with TV shows such as MasterChef, Bake-Off, etc... which engaged with the public and reinvigorated interest towards meal preparations."

*3.3. New Names and New Cuts*

Old culinary consumption practices faded away, and new ones came along. This dynamic sheds light on the transformation that affected the quotidian reality of butchers' artisanship. This change touches both the terminology of the trade as well as its materiality.

The second half of the 20th century marks the demise of dialects in Italy, the local Italic languages that were spoken by communities as their primary languages before the success of Italian as the primary language of Italy [52]. The interviewees indicated that the 1960s were the turning point of the process, after which the use of dialectal terms became less and less frequent, as customers started using Italian ones. An example is the muscles of the beef shoulder. This cut was locally called "punta du belin" (tr. tip of the dick). This term slowly lost use in favour of Italian alternatives. These could be more local terms, such as "sottopaletta" (tr. under slice) or "punta del codino" (tr. tip of the tail), or more widespread in Italy, such as "cappello del prete" (tr. priest's hat).

The lexical evolution did not stop, however. In the past 20 years, with the popularisation of international cooking broadcasting, such as MasterChef, as well as the success of international cuisines and the proliferation of international restaurants, such as Brazilian, Argentinian, and Peruvian ones, consumers became accustomed to and mesmerised by new recipes and preparations. This led to the use of terminologies borrowed from international culinary traditions, which thus led to the popularisation of the term "Picanha" to identify the cut.

The globalisation of taste and forms of meat consumption led the butchers to learn new terminology as well as new ways of cutting meat. In this respect, Int. 10′s shop is an example. There, the owner decided to marginalise the traditional cuts in favour of ones taken from the American BBQ tradition, including new products such as dry rubs (e.g., gin rubs, whisky rubs, rumshine rubs, truffle shuffle rubs, lemon pepper garlic rubs). Int. 10 has contacts all over the USA and managed to establish a "dictionary" for meat cuts with his colleagues, presented with the American definition and the Italian definition next to it. In this way, butchers from Italy are able to understand the request of customers and prepare the exact cut requested because, for example, "the top blade cut that could potentially become tenderloin, if it is cut horizontally, it becomes a cut called flat iron, and these are the only definitions established for these cuts", explains Int. 10.

Innovations in consumer practices led butchers to innovate their assortments. On the one hand, butchers inserted new products, as seen with the prontocuoci or the rubs of Int. 10. On the other hand, butchers experimented with new and alternative uses of the

traditional cuts. This is the case of the punta du belin used for BBQ. The same cut was the subject of other experimentations. Int. 1 reinvented its use by curing it and slicing away the fat from the cut in order to obtain raw meat to use for tartare or carpaccio. Similarly, Int. 8 works the neck, originally used for long-time cooking soups, to transform it into tartare, which is meat chopped with the knife. "Whether we like it or not", commented Int. 8, presenting his products and innovations, "we [butchers] have to evolve, even if this means that being a butcher today is definitely different from what it meant 20, 30, 50 years ago".

### 3.4. The Perceived Future

The need for change and a strong entrepreneurial capacity indicated by the respondents shows a perception of an uncertain future. Butchers predict increasing competition and a market jeopardised by new competitors, among which are supermarkets. While the overall consumption per capita of meat has almost tripled in 50 years (from 27 kg in 1961 to 79 kg in the 2010s; see Marchi, 2019), butcher shops turned out to be only one source of meat supply for urban consumers. "In the '80s, an Italian would buy around 43 kg of meat a year, whereas now we hardly reach 13 kg", highlighted Int. 2. However, consumers look at the traditional butcher shop for high-quality meat and refined meat products, which are not available in the other retail outlets, for which they are willing to spend more.

Additionally, the butcher shop retains the role of social venue, although having lost the centrality it still had in the 1970s. "In the '60s and '70s, there was a different routine adopted by the clients. They attended the 5:30 a.m. mass to then carry on to grocery shopping. We had different opening times and stayed open on Sundays because of the clients' routine. This is not relevant anymore, as people began to go to church less throughout the years. However, people come here in person. They like to physically purchase meat as an excuse to have a chat and shop simultaneously", explains Int. 8, "The relationship between the client and the artisan is the strong point of artisanship".

These assumptions—the importance of socialisation in the shop and the increasingly strong identification of butcher shops as places for (only) fine cuts—fuel the ongoing transformation of the shops. In the 1980s, with the introduction and growing space given to the sale of prepared cuts, butcher shops changed to assume a role closer to that of delicatessen shops. Nowadays, the changes have led to the restaurant sector. "In the 1990s, there was the first successful attempt to make a butcher shop as a small restaurant", remembers RC. Taking from the model of the American steak houses, since the first experiences of the 1990s, other butchers are experimenting with forms of catering that link the preparation of meat, its cooking, and service. "The future goes in that direction, unfortunately", remarks Int. 1, "The butcher's profession will become increasingly complex. It is no longer about opening up the shop in the morning and preparing the work table, selecting the meat, and serving it. It means the butcher of tomorrow should be an artisan, an entrepreneur, and a cook".

Alongside business complexity, the biggest uncertainty in the industry is the recruitment of new apprentices. "To this day there are not enough young adults in the full-time profession, and if there are, they are highly interested in the trends of butchery, rather than learning the profession from the bottom to the top", suggests Int. 5. The interviewees indicate as an example the trend of dry ageing: "They come here to acquire the knowledge on how to treat dry-aged meats rather than learning the basics. Thus, they are never become butchers, but just well . . . TikTokers!?!" (Int. 11).

"In the long run it is a profession that requires consistency and devotion, and the income at the beginning is quite mediocre. I had to give up a lot of my projects and dreams in order to keep up with this store", Int. 6 explains. "I had a young 20-year-old working with me and he was really good, but he felt like he was not moving forwards in this job, in comparison to the jobs that are offered today and the income they offer. You can understand that a 20-year-old wants to change path. The passion is fundamental; it's the fire that keeps you curious and going", explained Int. 10.

Recruitment concerns the interviewees also for the maintenance of their butchery knowledge. They highlight the substantial difference in the learning process and resulting knowledge carried out outside the shops. "When young apprentices are employed in supermarkets to work in the butchery section, they do not follow the same learning path as they do in a butcher shop. They learn to do one thing and only one thing, as in an assembly line. When a butcher teaches you the profession, he doesn't leave a single piece of information to do the job properly. Nothing is ruined, and you don't get hurt", Int. 4 explained. "Deboning is an example. Often in the butcher's section of the supermarket, we notice that certain cuts are not as fine as those found in butchers, for example, the amount of skin that remains on a certain cut of meat. If I leave a little skin on a cut, I'll be scolded by my master, and that's how it should be. This is not what happens there, where the first goal is to have ready things and full cabinets", Int. 12 remarked.

The model described by the informants focuses on the most extreme scenario in which the holistic approach that characterises artisan apprenticeship is substituted by a form of training in which knowledge is partitioned and its dissemination is functionalised according to the actual role of the employee in the chain of production [19]. The image reflects the archetypical distinction between artisan and factory worker, which is foundational to the contemporary understanding of artisanship [53]. Despite these limits, it conveys the fear concerning the loss of trade knowledge as well as one of the limitations of butchers' agency and entrepreneurship. In other words, it is the fear of marginalisation and decline. In the face of this perspective, some of the informants, such as Int. 7 and Int. 13, are planning public initiatives to "preserve" the profession in Genoa, such as establishing new training courses for young butchers who want to undertake the profession and teaching the old techniques and gastronomic knowledge underpinning the profession, otherwise perceived as lost.

Overall, the future of the trade appears to the butchers to be jeopardised, on the one hand, by the overall changes in consumption and, on the other, by the competition of the supermarkets and their approach to retailing. In this respect, butchers defend the distinction between their service and that provided by supermarkets as the only way forward, because "if butchers start competing with grand sale distributions, we lose at the start", says Int. 9. "The butcher of the future is as far as possible from the ideology of the supermarket because the butcher is an artisan and a shopkeeper, which is the dimension in which this profession gives its best", remarks Int. 10.

While the role of the shop as a place where clients find the best meat and also a place where they can acquire gastronomic knowledge and socialise is key, the butchers point out the need for them to learn from other sectors, primarily the restaurant sector, and expand the services offered by their shops: "We should go outside [the shop] because there is always something new to learn even though this is one of the most ancient professions in the world", explains Int. 9. The butchers also indicate the need to build a stronger relationship with breeders and the farming sector in general to reinforce the quality and strength of the supply chain, secure the quality of the materials, and learn about the supply chain to explain the cut of meat and the history behind it to the client.

## 4. Discussion

The research shows the change in the butcher's profession. The craft has become the subject of socioeconomic and cultural innovation faced with the profound transformations of consumption and purchasing practices, as well as the change in the areas in which meat retail is carried out.

The interviewees clearly indicate that the butcher's profession is an expression of embodied knowledge acquired through a process. This does not fully correspond to the model of peripheral legitimation proposed by Lave and Wegner [54] and repeatedly taken up in the scholarship on artisanship, e.g. [14,24,55]. Lave and Wenger [54] have described learning a craft as a contextual social process that involves the trainee's engagement with a community of practitioners. Through this form of quotidian interaction, a trainee learns

expert knowledge, skills, and competences and is legitimised by the other professionals. By performing increasingly complicated tasks and interacting with the others, the trainee learns the trade but, above all, embodies the vocabulary, values, and organising principles shared by the professional community. The trainee acquires the legitimacy and prestige required to be considered a full-fledged professional both within the community of practitioners and within the wider context of the society. Thus, where this model places the acquisition of prestige within the community of practitioners at the centre of professional growth, the butchers suggest that individual prestige takes on a secondary role, while great emphasis is placed on the individual's ability to acquire techniques, correctly apply them, and potentially innovate them. In this respect, what the butchers describe appears a path more traditionally associated with vocational training [56], which is built around the acquisition and application of professional know-how. Butchery does not encompass only gastronomic know-how; it also involves commercial and management know-how. This configuration expands the traditional understanding of professional artisanship to include the economic and entrepreneurial aspects. In so doing, it makes artisanal knowledge fit the French expression "savoir, savoir-faire, et faire savoir" (to know, to know how to do, to do what others know). This knowledge is embedded within the person of the artisan; however, the research suggests the artisan is always embedded and can be understood only in the framework of the microcosm of the shop.

The shop is the silent protagonist of the stories of the butchers. In the shop, they learn butchery and practice it. The shop is also the place of contact between the artisan and the world and thus the stage where they perform their artisanship and through which they can affirm the sociocultural status of their profession, as Spinelli [33] illustrates. The shop, however, is also the expression of the economic and social success of the individual butcher, who finds material expression in the increasing number of customers, the volume of business, and the size and fame of the venue. The shop is so important and central in the narrative that a butcher outside this space is no longer recognised by his colleagues as a proper butcher. As the descriptions of butchers employed in supermarkets show, a butcher without a shop is not recognised for his professional status, and his skills are described as partial or limited in their expression.

The interviews outline two main innovation dynamics of the butcher shop. On the one hand is the dynamic properly linked to the evolution of the forms of what Pye famously defined as the workmanship of risk, which is any "workmanship using any kind of technique or apparatus, in which the quality of the result is not predetermined, but depends on the judgment, dexterity and care which the maker exercises as he works" [57]. The interviews show this ability has evolved over the last few decades. The field of expression and application of artisanship expanded and shifted, experimenting with new preparations (e.g., the prontocuoci), drawing on and hybridising with different traditions of butchery, and taking from other sectors, such as the catering sector, to empower and revamp the business. In this sense, food artisanship becomes adaptable to the socioeconomic context in which it is expressed. The displacement of the field of practice brings with it a change of language, retail practices, business management forms, and relations with the city and the field of commerce. Precisely because of this constant dialogue with the contingent, it can be said that craftsmanship is the daughter and expression of a particular historical and geographical assembly [58], always current and constantly changing.

On the other hand, the transformation is linked to the shops and their size, maintenance, and vitality. The transformation outlines a slow erosion of the sociocultural and economic centrality of the shops in the city. This shrinks, as we have seen, the very space where the butchers affirm their social status. The present commercial dynamics compress it in an increasingly limited space: in a red ocean [59] where the chances of economic success are weathered. Given the expectations of modernity [60], the hope for success felt by the butchers is obfuscated, forcing them to innovate their business to maintain their position. This condition affects the vision of the future. It becomes uncertain from an economic point of view and especially from the point of view of the continuity of the profession.

The absence of a new generation of apprentices, who have moved away from the profession in light of the difficulties and uncertainty, strengthens the sense of uncertainty and precariousness, causing butchers to question the sociocultural significance of their art.

## 5. Conclusions

The configuration of dynamics that this research has explored highlights the historical and territorial sense of gastronomic artisanship. It suggests reading the artisan techniques and practices as cultural expressions always embedded in a specific economic and food reality. At the same time, the research shows that the cultural significance of artisanship is not limited to mere food preparations but must always be read in its socioeconomic dimension. The research suggests that artisanship should be read as a form of entrepreneurship capable of placing and moving the craftsman within the global social hierarchy of a community [24].

Therefore, the preservation of artisanship and its sociocultural complexity cannot be limited to the mere preservation of gastronomic forms and techniques. While training and Vocational Education and Training programmes aimed at the continuation of butchery's know-how are becoming increasingly popular, the significant aspects concerning small and family entrepreneurship are marginalised. This leads to the emptying of artisanship of the very worldview that underpins it, reducing artisanship to mere technique [61].

While the ongoing socioeconomic processes seem to lead to the closing of butcher shops and all other small shops that are the basis of proximity commerce, the preservation of artisanship as a form of entrepreneurship can support the maintenance of the forms of local commerce and thereby the vitality of urban centres. The case study of Genoa raises questions and opens the ground to new research about how to fully support the continuation of artisanship in urban contexts deeply touched by modernisation and ongoing globalisation.

**Author Contributions:** Conceptualization, M.F.F. and A.C.; methodology, M.F.F.; validation, M.F.F.; formal analysis, M.F.F.; investigation, A.C.; resources, M.F.F. and A.C.; data curation, A.C.; writing—original draft preparation, M.F.F. and A.C.; writing—review and editing, M.F.F.; visualization, M.F.F.; supervision, M.F.F.; project administration, M.F.F.; funding acquisition, M.F.F. All authors have read and agreed to the published version of the manuscript.

**Funding:** The research is part of the project NODES, which has received funding from the MUR-M4C2 1.5 of PNRR with grant agreement no. ECS00000036, specifically for what concerns the modelling of the socio-cultural dynamics concerning the ending part of meat agrifood chain considering how Piedmontese animals (such as cows, chickens and pigs) are commonly traded and sold in Genoa, making the dynamics of the local market relevant for understanding the possibility of innovation for the entire meat sector of NW Italy. The APC were fully waivered by MDPI.

**Institutional Review Board Statement:** The study was conducted in accordance with the Declaration of Helsinki, and the AAA's Anthropological Ethics Guidelines.

**Informed Consent Statement:** Informed consent was obtained from all subjects involved in the study.

**Data Availability Statement:** Relevant data were included in the text. Full data set available in: Costa, A. *Butchery in Genoa.* Thesis (BA), University of Gastronomic Sciences, 2022.

**Conflicts of Interest:** The authors declare no conflict of interest.

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
