# Peer review of "Keeping the Knives Sharp: Socioeconomic Innovation in the Artisan Sector of Butchery in Italy"

_societies, doi:10.3390/soc13040080_

Round 1

Reviewer 1 Report

This is a very well written, researched and presented study of the socio-cultural significance of artisan butchery in Genoa in the context of the receding economic centrality of the shop.

The study is based on interviews which very well illustrate the points the authors are making about the socio-cultural continuity and discontinuities in artisan butchery. 

A novel contribution in studying the artisan aspects of butchery and its cultural significance, especially in culturally conservative regions. It would be interesting to see how this artisanry responds to multiculturalism in superdiverse urban contexts but this is obviously not the purpose of this study.

Author Response

Thank you to the reviewer and the sustain given to our research

Reviewer 2 Report

In this manuscript the authors report a detailed ethnographic investigation into small-scale butcher shops in Genoa in order to investigate a particular form of food-related artisan practice.  The manuscript is well- and clearly written and the argument is well presented and relevant to food studies.  I believe this paper is appropriate for publication with only minor revisions. 

In line 162, the authors state without much discussion "Butchery is an art", but given the detailed focus on the nature of artisanship I feel this statement may be a bit fraught.  Is it an art?  A craft?  An embodied practice?  I think that this statement is meant as merely a transition, but given the nature of the manuscript I found it jarring.

In line 274 I believe the authors meant to translate the term as "delicacies" rather than "delicatessen", which is an English metonymic adoption of an (I assume!) German term that perhaps holds that original meaning.

In lines 438-440 the authors hold their findings in contrast to the theories of artisanship proposed by Lave and Wegner, but as far as I can tell the authors don't really describe these theories at any point in the manuscript.  Since this seems to be a major thrust of the argument, I believe the manuscript would be improved if the authors described these theories and demonstrated how their findings are in disagreement with them.

Author Response

Thank you to the reviewer.

We accepted all the comments, by correcting art into craft and delicatessen into delicacies. Moreover, a summary of the theory of Lave and Wegner was introduced (lines 440-448).